# Contrastive Learning on Synthetic Videos for GAN Latent Disentangling

**Kevin Duarte**
Adobe Inc. (ASML)
kduarte@adobe.com

**Wei-An Lin**
Adobe Inc. (ASML)
wlin@adobe.com

**Ratheesh Kalarot**
Adobe Inc. (ASML)
kalarot@adobe.com

**Jingwan Lu**
Adobe Research
jlu@adobe.com

**Eli Shechtman**
Adobe Research
elishe@adobe.com

**Shabnam Ghadar**
Adobe Inc. (ASML)
ghadar@adobe.com

**Mubarak Shah**
Center for Research in Computer Vision
University of Central Florida
shah@crcv.ucf.edu

## Abstract

In this paper, we present a method to disentangle appearance and structural information in the latent space of StyleGAN. We train an autoencoder whose encoder extracts appearance and structural features from an input latent code and then reconstructs the original input using the decoder. To train this network, we propose a video-based latent contrastive learning framework. With the observation that the appearance of a face does not change within a short video, the encoder learns to pull appearance representations of various video frames together while pushing appearance representations of different faces apart. Similarly, the structural representations of augmented versions of the same frame are pulled together, while the representation across different frames are pushed apart. As face video datasets lack sufficient number of unique identities, we propose a method to synthetically generate videos. This allows our disentangling network to observe a larger variation of appearances, expressions, and poses during training. We evaluate our approach on the tasks of expression transfer in images and motion transfer in videos.

## 1 Introduction

In recent years, Generative Adversarial Networks (GANs) have become effective in generating high-quality samples. Among them, StyleGANs [1, 2, 3, 4] have demonstrated amazing ability to produce high-definition and photo-realistic face images. The fact that StyleGANs induce a semantically interpretable latent space has enabled many downstream face editing tasks [5, 6, 7, 8, 9, 10, 11, 12, 13]. In this work, we aim to discover a disentanglement which separates appearance and structural features in the latent space. Here, we define **appearance features** as subject-dependent properties including face identity, hair style, skin tone, eye color, facial hair, presence of glasses, etc., and **structural features** as subject-independent properties like face pose and expression. Unlike common facial attributes such as smile, lip color, or head pose that can be characterized using a small number of latent dimensions [8], face appearance features (e.g. face identity) are difficult to measure and often heavily correlated with structural features. To tackle this problem, we hypothesize that within a short video clip, since the variations of subject-dependent properties are smaller compared to subject-independent properties, the "appearance features" extracted from the latent codes of individual frames should be close to each other. We propose to realize this idea by training an autoencoder in the StyleGAN latent space using a novel *video-based latent contrastive learning* so that the encoder extracts appearance and structural features from input latent codes, and the decoder reconstructs the original input.

Contrastive learning has been shown to be effective in visual representation learning [14, 15, 16, 17, 18] and unpaired image-to-image translation [19]. The idea is to encourage the networks to

NeurIPS 2022 Workshop on Synthetic Data for Empowering ML Research.

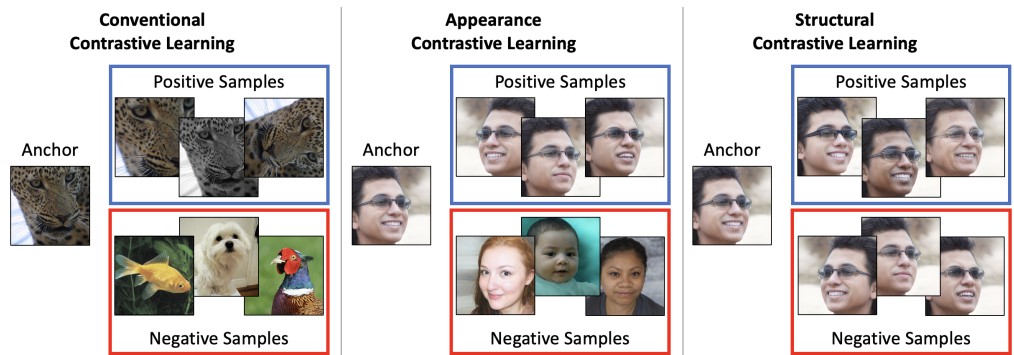

Figure 1: Comparison of our video-based latent contrastive learning approach with conventional contrastive learning for self-supervised learning [14]. Conventional contrastive learning uses augmented versions of the anchor image as positive samples and different images as negatives. Our proposed appearance contrastive loss uses different frames of the same video as positive samples and different identities as negative samples. The structural contrastive loss uses visually augmented versions of the anchor image as positives and different frames of the video as negatives.

associate visual cues at image/patch level by training on a large number of corresponding and non-corresponding images/patches. The corresponding pairs can be defined as images augmented by simple color transformation or patches in the same spatial location [19]. In StyleGAN latent space, however, finding a large number of corresponding and non-corresponding pairs through simple image transformation is challenging since the association between image space operations and StyleGAN latent representation is non-trivial or even intractable. In this work, we consider those pairs discovered by the latent representation of video frames. Specifically, for appearance features, different frames within the same videos are considered corresponding pairs and video frames from different subjects are non-corresponding pairs. For structural features, video frames with similar expression and pose are considered as corresponding pairs whereas different frames within the same video are non-corresponding pairs. In Figure 1, we present the core differences between conventional contrastive learning and the proposed video-based latent contrastive learning.

Successfully learning with a contrastive objective requires a large number of face videos such that there is sufficient variation of appearance, expressions, and poses. However, most face video datasets do not contain enough unique identities to learn robust appearance and structural representations. We propose to exploit the property that video frames embedded in StyleGAN latent space are intrinsically low-rank, and the latent codes can be easily augmented by projecting into the subspace spanned by these frame embeddings. By augmenting the latent codes, we generate synthetic video frames to ensure sufficient number of unique identities with different expressions and poses. To our knowledge, the use of such a contrastive objective to disentangle GAN latent representations and this manner of generating synthetic videos for training has not been previously explored.

## 2    Related Work

**Disentangled representation learning** One of the main goals in representation learning is to disentangle the underlying factor of variations that explain real-world data. Most existing works aim to disentangle factors such as background, object, shape, or texture [20, 21, 22]. In unsupervised image-to-image translation [23, 24, 25, 26], disentanglement between content and style is commonly adopted to achieve *global* texture transfer between multiple domains. On the other hand, our formulation of learning disentangled appearance and structural features focuses on a more fine-grained disentanglement. Peng *et al.* [27] propose disentangling "identity" and "non-identity" features in a supervised manner from samples generated by a 3DMM. Recently, Nitzan *et al.* [28] train a deep CNN to disentangle the *identity* and other facial *attributes* (pose, expression, and illumination) with a cyclic and adversarial objective. In contrast, our approach is applied directly on the GAN latent space ($w \in \mathcal{W}+$) and our video-based contrastive learning objective is novel for this task.

**StyleGAN-based video generation** Earlier works on GAN-based video generation are mainly limited to generating videos in lower resolution. Recently, StyleGAN's capability of generating photo-realistic images has motivated some work to synthesize short video clips by traversing in the StyleGAN latent space [29, 30, 31]. In [29], the authors propose to synthesize smooth random motion

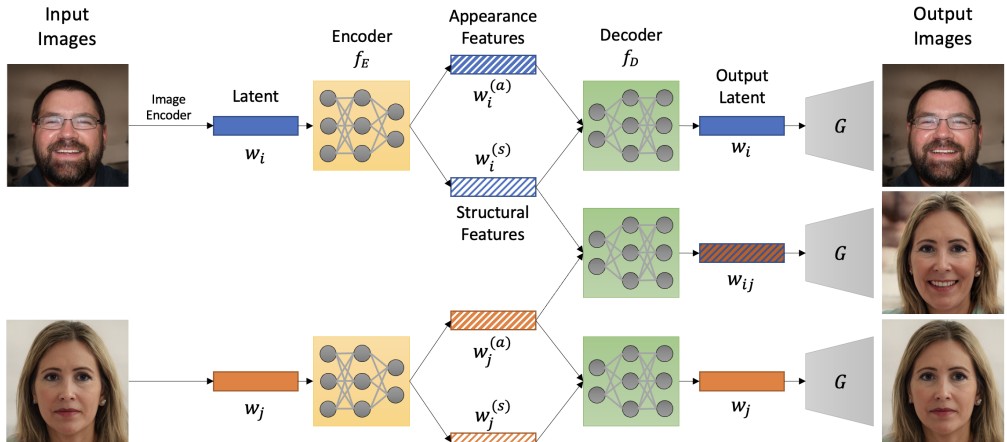

Figure 2: Network Architecture Overview. Given an input latent code, our encoder disentangles the appearance and structural features. These disentangled features are passed to the decoder to reconstruct the input latent code. When appearance and structural features from different images are input to the decoder, a new face is generated with the appearance of one face and the pose and expression of the other.

on user-specified facial regions from a single image. In [30, 31], it has been shown that temporal dynamics in a video clip can be learned by a latent space generator. However, this requires training one latent generator for each video clip. In contrast, our work can synthesize new videos with the same motions as the driving video without the need to re-train the network at inference time.

## 3   Our Approach

The goal of this work is to disentangle the GAN latent space into appearance and structural features. Once disentangled, features from different images can be swapped to perform tasks like expression and motion transfer. To this end, we train a latent autoencoder (AE) with an encoder, $f_E$, and a decoder, $f_D$, so that given a latent code $w \in \mathcal{W}+$, we obtain $w^{(a)}, w^{(s)} = f_E(w)$, where $w^{(a)}$ captures appearance and $w^{(s)}$ captures structural features. The decoder reconstructs the input latent from these disentanged representations, $w = f_D\left(w^{(a)}, w^{(s)}\right)$. This is depicted in Figure 2. For training, we assume the existence of $K$ video sequences $\{v_k\}_{k=1}^K$, such that the $T_k$ frames in each video (where $T_k$ can vary between videos) is encoded into latent vectors, $\{w_{k,t}\}_{t=1}^{T_k}$. Next, we present a video-based contrastive objective to train $f_E$ to disentangle appearance and structural features.

### 3.1   Video Contrastive Learning

**Contrastive Appearance Loss**   Under the assumption that a person's appearance does not change throughout a video sequence, we attempt to push appearance representations of frames within the same video to be closer to each other than the appearance representations across different videos. That is, $|w_{i,t}^{(a)} - w_{i,t'}^{(a)}| < |w_{i,t}^{(a)} - w_{j,t''}^{(a)}|, \quad i \neq j$. This can be accomplished using a contrastive loss [32, 14], where positive samples are two frames taken from the same video and negative samples are frames taken from different videos. Formally, for a batch with $B$ samples we use the Masked Margin Softmax (MMS) loss [33], defined as

$$\mathcal{L}_a = \frac{1}{B} \sum_{i=1}^{B} \frac{e^{S\left(w_{i,t}^{(a)}, w_{i,t'}^{(a)}\right) - \gamma}}{e^{S\left(w_{i,t}^{(a)}, w_{i,t'}^{(a)}\right) - \gamma} + \sum_{j \neq i} e^{S\left(w_{i,t}^{(a)}, w_{j,t''}^{(a)}\right)}}, \tag{1}$$

where $S$ is a similarity function between two vectors and $\gamma$ is a margin hyper-parameter. Here, we use the negative L2 distance, $S(a, b) = -\sum(a - b)^2$, as the similarity measure.

**Contrastive Structural Loss**   We learn the structural features in a similar manner. In this case, positive samples would consist of different subjects with the same expression and pose, and negative

samples would consist of the same face with different expression and pose. Although it is simple to obtain the negative samples by sampling different frames of the same video, obtaining two faces with the same expression would require manual annotations. To this end, we augment versions of the same face to obtain these pairs as described in 3.2. The augmented version of a latent code is denoted $\tilde{w}_{k,t}$. This produces the appearance and structural features, $\tilde{w}_{k,t}^{(a)}$ and $\tilde{w}_{k,t}^{(s)}$ respectively. Therefore, the structural contrastive loss is defined as

$$\mathcal{L}_s = \frac{1}{B} \sum_{i=1}^{B} \frac{e^{S\left(w_{i,t}^{(s)}, \tilde{w}_{i,t}^{(s)}\right) - \gamma}}{e^{S\left(w_{i,t}^{(s)}, \tilde{w}_{i,t}^{(s)}\right) - \gamma} + \sum_{t' \neq t} e^{S\left(w_{i,t}^{(s)}, w_{i,t'}^{(s)}\right)}}. \tag{2}$$

## 3.2 Training Data Generation

**Video Generation**   Since contrastive learning often requires large amounts of data, existing datasets often contain insufficient unique identities to learn robust appearance and structural representations (*e.g.* the VoxCeleb2 dataset [34] contains only 5992 individuals). Therefore, we propose a method to generate a large amount of synthetic videos which have variation of appearance, expressions, and poses. Fox *et al.* [30] propose the *Offset Trick* to transfer the motion of a generated video to different subjects. Given a real video, we project each frame into the GAN latent space and use the Offset Trick to transfer the motion to multiple sampled latent codes' generated faces.[1] With a sufficiently large number of sampled latent codes, we can ensure our network trains on videos which contain a greater variation in appearance. Specifically, we train our AE on a large synthetic video dataset by randomly sampling 5 million latent codes and using the Offset Trick on 1000 VoxCeleb2 videos.

**Image Augmentation**   To obtain positive samples for the contrastive structural loss, we augment the face images in a video using style-mixing [1] and latent space sliders [7]. For style mixing, we randomly mix latent codes from the last 10 styles (layers 8-18) in the $\mathcal{W}+$ latent space. We further augment faces by randomly traversing the latent space in the direction of controllable sliders obtained by [7]. We select sliders that correspond with appearance and do not change pose or expression: "age", "bald", and "skin tone". Combining these sets of augmentations allows for a change in the face's appearance while ensuring their pose and expression remain the same. Moreover, we increase the variety of poses and expressions by including the horizontally flipped version of faces during training. These flipped images not only improve the learned structural representations, but also are useful "hard positives" for the appearance contrastive loss: a horizontal flip is a more drastic head pose change than what is present in most videos.

## 3.3 Auxiliary Losses

**Latent and Image Cyclic Losses**   We include cyclic losses to to ensure the encoder produces consistent appearance and structural features, even when the input latent is obtained from two different images. For latent codes $w_i$ and $w_j$, we extract the appearance and structural features $w_i^{(a)}$ and $w_j^{(s)}$. Then, these features are given to the decoder network to obtain a new latent code $f_D(w_i^{(a)}, w_j^{(s)}) = w_{ij}$. If the encoder is able to extract disentangled appearance and structural features, it is expected to recover $w_i^{(a)}$ and $w_j^{(s)}$ from $w_{ij}$. Therefore, with the resulting disentangled features $f_E(w_{ij}) = \hat{w}_i^{(a)}, \hat{w}_j^{(s)}$, we can compute the a latent cyclic loss $\mathcal{L}_{cyc-lat} = \left(w_i^{(a)} - \hat{w}_i^{(a)}\right)^2 + \left(w_j^{(s)} - \hat{w}_j^{(s)}\right)^2$ for B pairs of latent codes in the batch. Similarly, we compute an image-based cyclic loss which is computed on image pixels as opposed to latent codes. Here, the GAN generator, $G$, generates an image from the latent code output from the decoder network. The image cyclic loss is calculated as $\mathcal{L}_{cyc-img} = \left[x_i - G\left(f_D\left(\hat{w}_i^{(a)}, w_i^{(s)}\right)\right)\right]^2$, where $x_i$ is the original input image.

**3DMM Consistency Loss**   Lastly, we include a 3D Morphable Model based loss which improves the quality of the disentangled features. Given an image, CNNs can be used to regress various coefficients (pose, expression, identity, texture, and lighting[2]) of a 3DMM face model. We employ a

---

[1] Identities can change when there are large pose variations, but in the majority of cases, the face pose only has minor changes throughout a video. Refer to supplemental materials for examples of this behaviour.

[2] In this work, we do not include lighting coefficients during training.

Table 1: Results on Motion Transfer on the VoxCeleb dataset.

| Method | AKD↓ | ACED↓ | FID↓ | FVD↓ |
|---|---|---|---|---|
| Offset Trick [30] | 2.164 | 0.771 | 121.0 | 272.8 |
| FOMM [43] | 2.388 | 0.701 | 148.1 | 244.8 |
| Ours | 1.928 | 0.711 | 139.5 | 275.8 |

pretrained model [35] to estimate these coefficients during training, and enforce consistency across faces generated with the same appearance and structural features. Images generated from the same appearance features should have consistent identity and texture coefficients. Meanwhile, the pose and expression should be consistent across images with the same structural features. Let us denote $M_a(x)$ as the regressed appearance coefficients (identity and texture) and $M_s(x)$ as the regressed structural coefficients (pose and expression) for an image $x$. We then define the losses between pairs of images $(x_i, x_j)$ as $\mathcal{L}_{3DMM_a} = [M_a(x_i) - M_a(G(w_{ij}))]^2$ and $\mathcal{L}_{3DMM_s} = [M_s(x_i) - M_s(G(w_{ji}))]^2$.

### 3.4 Network Architecture

The encoder, $f_E : \mathbb{R}^{18 \times 512} \to \mathbb{R}^{d_a} \times \mathbb{R}^{d_s}$, consists of two MLPs which output the appearance and structural features, with dimensions $d_a$ and $d_s$, respectively. The decoder, $f_D : \mathbb{R}^{d_a} \times \mathbb{R}^{d_s} \to \mathbb{R}^{18 \times 512}$, consists of a three layer MLP. It combines these disentangled features and outputs a latent representation that can be used by the GAN to generate an image. As this is an autoencoder, both the encoder and decoder are trained jointly using a reconstruction loss: $\mathcal{L}_{rec} = [w_i - f_D(f_E(w_i))]^2$. The network is trained end-to-end using a weighted sum of reconstruction, contrastive, cyclic, and 3DMM consistency losses. At inference time, this network can be applied to expression transfer by passing the appearance features of one face image and the structural features of the target expression image through the decoder. Similarly, motion transfer is performed by using the appearance features of the source image and the structural features from each frame of the driving video.[3]

## 4 Experimental Evaluation

**Implementation Details and Evaluation Protocol** As described in Section 3.1, we train on a large synthetic video dataset, generated from a 1000 video subset of the VoxCeleb2 dataset [36, 34] and 5M randomly sampled latent codes. To obtain GAN latent codes from images and to generate images from a given code, we use a pre-trained e4e [37] and StyleGAN2 [2] respectively. We quantitatively evaluate our method on the task of motion transfer. Given a driving video and a source image, the driving video's motion is transferred to the source image to obtain an output video. We report the Average Keypoint Distance (AKD), Average Classifier Embedding Distance (ACED), Frèchet Inception Distance (FID) [38] and Frèchet Video Distance (FVD) [39] metrics. AKD uses a pretrained facial landmark detector [40] and measures the distance between landmarks in the output and driving videos; ACED measures the $\mathcal{L}_1$ distance between the embedding layer of a ResNet-50 [41] face classifier trained on the UMDFaces dataset [42]. The ACED and AKD metrics should be viewed jointly: whereas ACED measures how well a method maintains a person's identity, AKD measures how well pose and expression are preserved. FID and FVD are used to evaluate each methods' image naturalness and motion quality, respectively. Additional information on hyper-parameters and evaluation protocol can be found in the supplement.

### 4.1 Quantitative Evaluation

We present a comparison with two previous methods [43, 30] in Table 1. We first compare with the Offset Trick [30], which assumes that the directions in $\mathcal{W}+$ space which control identity and motion (*i.e.* expression, pose, and articulation) are mostly orthogonal. Our method outperforms the Offset Trick on both AKD and ACED metrics. Generally, the Offset Trick transfers motion well, but tends to fail when there are large pose differences between the source image and frames in the driving video. Furthermore, the identities produced by the Offset Trick can shift throughout the output video depending on the changes in pose of the driving video. In contrast, our method tends to maintain a consistent identity for all frames of the output video.

Next, we compare with the first order motion model (FOMM) [43], which estimates a dense motion field from a driving frame to the source image and passes this motion field to a generation module

---

[3]More details about the inference process is included in the Supplementary Materials.

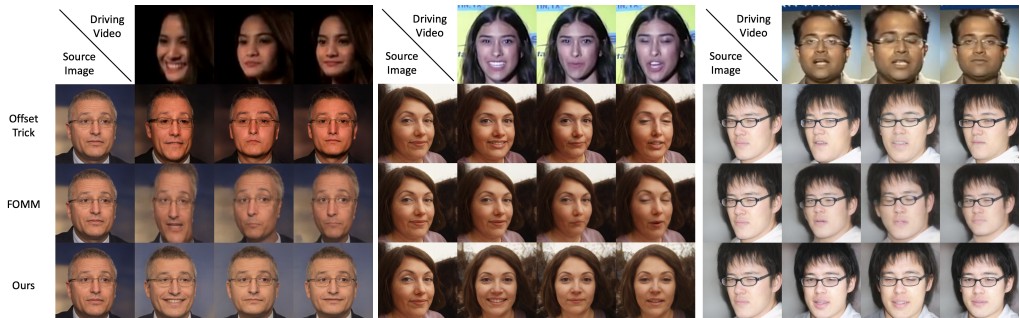

Figure 3: Qualitative comparisons with theOffset Trick and First Order Motion Model (FOMM) for the task of motion transfer. Here, the motion of a VoxCeleb video is transferred to an FFHQ image.

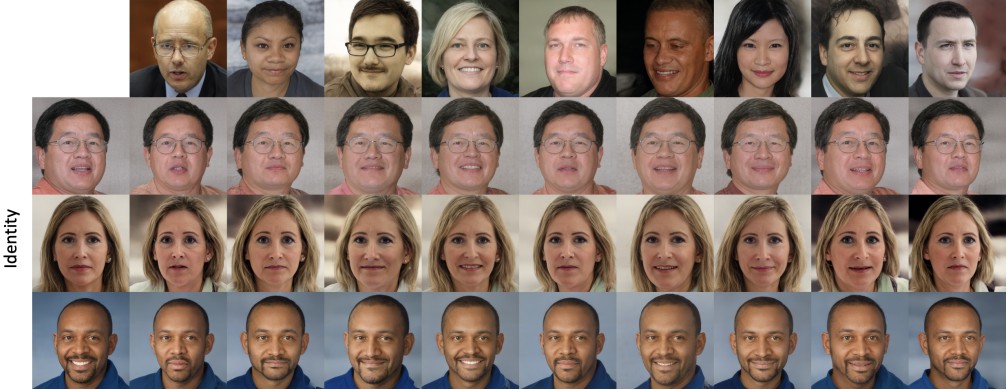

Figure 4: Expression transfer examples of our method. The first row contains the various expressions which will be used, and the first column in rows 2-4 contains the identities. We find that the identity is maintained across different expressions.

which warps the source image to generate the output frames. FOMM slightly outperforms our method on the ACED metric. Although FOMM outputs faces which are similar to the source image, the generated images tend to have much lower quality; not only do they have a lower resolution (256x256 when compared with the StyleGAN's 1024x1024), but also the warping operation may generate deformed faces. This artifact is especially prevalent when there are sudden facial movements or when the source face's pose greatly differs from the poses present in the driving video, and could explain the undesirable increase in the AKD and FID scores.

## 4.2 Qualitative Evaluation

**Motion Transfer**    In Figure 3, we present motion transfer examples using the Offset Trick, FOMM, and our proposed approach. Here, the motion for VoxCeleb videos are transferred to FFHQ [1] images. Although the Offset Trick tends to mimic the the driving videos' facial expression changes, there is often a change in the identity throughout the output video. The FOMM preserves the identity of the actor throughout the video, but it generates lower resolution frames and faces which seem warped, as depicted in the figure's first example. In the supplemental materials, we include a variety of videos generated by our method.

**Expression Transfer**    Our method can also perform expression transfer on individual images. Since we combine the structural features of one image and the appearance features of another, the resulting image has the expression and pose corresponding to the prior image. We present examples of our method performing expression transfer between FFHQ images in Figure 4. The generated faces maintain identity across a variety of target expressions.

## 5   Conclusion

We have presented a novel approach to disentangle GAN latent representations using a video-based contrastive objective. The latent codes are decomposed into appearance and structural features. Appearance features represent those aspects of the face which remain consistent throughout a video, such as identity, hair-style, and presence of glasses; structural features represent those aspects that

change, like expression and face pose. To train our network, we proposed a method for synthetically generating a diverse set of video frames by augmenting a small set of real video with randomly sampled latent codes. Once trained, our method can perform various tasks like expression transfer in images and motion transfer in videos.

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
