# OpenReview forum: "Contrastive Learning on Synthetic Videos for GAN Latent Disentangling"
_NeurIPS.cc/2022/Workshop/SyntheticData4ML — Neurips 2022 SyntheticData4ML_

### Official Review · Reviewer_t9dx · 2022-10-17
**The organization of this paper is clear and easy to understand. More discussion on the limitation of this method is needed when comparing with FOMM and Offset trick. Overall, it demonstrates decent innovation and use of techniques.**

**Rating:** 6
**Confidence:** 4

**Review:**

This work provide clear description on how to construct positive and negative samples pairs for disentangling appearance and structural components. Extensive evaluation shows pros and cons among relative approaches. In particular, the method for generating synthetic videos would be of interest for related researches where large amount of training data is needed. Finally, they demonstrate the advantage of the proposed approach on motion and expression transfer as compared to existing frameworks.

**Clarity**:
The use of words in Introduction should be polished to reflect the novelty of this work. For example, the last paragraph of Introduction needs to be succinct and highlight the key innovation.

**Pros**:
This work is interesting as it extends contrastive learning based on synthetic videos to produce decoupled appearance and structural features. GANs have demonstrate faithful generation of high-quality images. This study provides a route to achieve latent disentangling. It would be of interest for applications where it requires controlled synthesis of natural images or video.

The authors combine synthetic video frames with augmented techniques to generate sufficient training samples. This approach produces diverse identities with varied expressions and poses.

**Cons**:
1. As mentioned in supplementary materials, 50 driving videos from VoxCeleb2 in varied lengths are utilized to evaluate. I feel this evaluation dataset is very small, which could lead to biased results. In addition, does the length of videos used for evaluation impact reported metric scores?

2. This proposed method does not generalize well when there are significant differences between source image and driving videos. I think authors should clearly point this out in manuscript with in-depth analysis. Why does the failure case occur and how potential solutions could be examined?

3. After examining the videos attached in the supplementary materials,  my observation is: while First Order Motion Model (FOMM) suffers from poor image/video resolution, it exhibits better fluency and expression naturalness as compared to proposed method. Since the goal of this work is focused on generating synthetic videos, it would be helpful for future readers if authors can incorporate additional discussion on this aspect.

---

### Official Review · Reviewer_iGU5 · 2022-10-17
**In all, the contributions of the technique are clearly stated in the paper. The paper is clear, easy to follow and well written for each section.  But it spent a lot of space to discuss methods,  and experiment details are lacking a bit. It could be better if method limitation is discussed more.**

**Rating:** 6
**Confidence:** 5

**Review:**

Authors are proposing a method to disentangle appearance and structural information in the latent space of StyleGAN.  They train an auto-encoder whose encoder extracts appearance and structural features from an input latent code and then reconstructs the original input using the decoder. As data is limited for training, it also propose a method to synthetically generate videos. The approaches were evaluated on tasks of expression transfer in images and motion transfer in videos.
The results show that it outperform Offset trick in pose/expression preserving and identity preserving. The method is not as well as FOMM in preserving identity. However, FOMM is lacking in naturalness and motion quality. So the authors still conclude the proposed approach is better than both Offset trick and FOMM when it comes to motion transfer on VoxCeleb dataset

In all, the contributions of the technique are clearly stated in the paper. The paper is clear, easy to follow and well written for each section.

Pros:
The proposed loss is comprehensive. Besides the constructive loss in structural and appearance. It covers aspects both in latent space and image space for cyclic loss. It also includes a 3DMM consistency loss that improves the quality of the disentangled feature via comparing identity/texture , pose/expression.
It leverage the same video frames to work as positive samples for the appearance constructive learning, while noticed that hard to get positive data for structural constructive learning, so it turns to synthetic images for positive samples from anchor image just to preserve the pose and expression, but vary the other dimensions like identities.

Cons:
It could be better if it explore more about the limitation of the techniques.
It could be better if more experiments/ablation study is done.
Too many losses it is hard to adjust the coefficient and balance each other in training
Less training details are covered in the paper

---

### Official Review · Reviewer_jfmP · 2022-10-18
**Good combination and applications of tools, however weak baselines and related work to estimate impact**

**Rating:** 6
**Confidence:** 4

**Review:**

* The authors explore the use of contrastive losses to disentangle the learning of structural and appearance representations, furthermore, the authors take inspiration from existing approaches to generate additional data to utilize for contrastive learning.
* Overall, the paper is well written and clearly structured, however, there could be additional quantitative and qualitative investigations to understand the impact of each component.
* The lack of strong baselines also makes it hard to evaluate the impact of the various components proposed in the paper.
* The authors also not explore the use of the end-to-end model without any additional supervision such as the pre-trained model used for the 3DMM consistency loss.
* Overall, the related work section and the baselines discussed are relatively weak and are the lagging sections in the paper. Strengthening them would result in a much stronger contribution.
* Overall the paper aligns well with the workshop theme and demonstrates the utility of using synthetic data to supplement such approaches and improve performance.

---

### Meta-Review · Area_Chair_kT6E · 2022-10-19

**Recommendation:** Accept